# Pectin in Metabolic Liver Disease

**DOI:** 10.3390/nu15010157

**Published:** 2022-12-29

**Authors:** Wanchao Hu, Anne-Marie Cassard, Dragos Ciocan

**Affiliations:** 1Faculté de Pharmacie, Université Paris-Saclay, Inserm U996, Inflammation, Microbiome and Immunosurveillance, Bâtiment Henri MOISSAN, 17 Avenue des Sciences, 91400 Orsay, France; 2Paris Center for Microbiome Medicine (PaCeMM) FHU, 75011 Paris, France; 3AP-HP, Hepatogastroenterology and Nutrition, Hôpital Antoine-Béclère, 92140 Clamart, France

**Keywords:** gut microbiota, pectin, fiber, ALD, NAFLD, alcoholic hepatitis, bile acids, AhR, indoles

## Abstract

Alterations in the composition of the gut microbiota (dysbiosis) are observed in nutritional liver diseases, including non-alcoholic fatty liver disease (NAFLD) and alcoholic liver disease (ALD) and have been shown to be associated with the severity of both. Editing the composition of the microbiota by fecal microbiota transfer or by application of probiotics or prebiotics/fiber in rodent models and human proof-of-concept trials of NAFLD and ALD have demonstrated its possible contribution to reducing the progression of liver damage. In this review, we address the role of a soluble fiber, pectin, in reducing the development of liver injury in NAFLD and ALD through its impact on gut bacteria.

## 1. Introduction

Alcohol abuse and overweight/obesity are the two main causes of liver disease in western countries, with no therapeutic options in the early stages, other than losing weight and alcohol withdrawal, and very few in the advanced stages of the disease [1,2]. Nutritional liver diseases, including alcoholic liver disease (ALD) and non-alcoholic fatty liver disease (NAFLD), share common histopathological features: upon exposure to the deleterious stimuli (alcohol or a western diet enriched in fat and/or simple sugars) lipids accumulate in the liver, a condition called steatosis, that can be accompanied by episodes of inflammation (alcoholic hepatitis and steatohepatitis or non-alcoholic steatohepatitis (NASH)), leading to fibrosis and, ultimately, cirrhosis and liver cancer [3,4]. A consensus of international experts has proposed that the disease acronym be changed from NAFLD to metabolic dysfunction-associated fatty liver disease or ’MAFLD’ to more accurately reflect pathogenesis and better integrate the current understanding of patient heterogeneity [5]. However, as the studies cited in the present review mostly used the term NAFLD, we will use this acronym throughout the present paper. The global burden of these two diseases is increasing worldwide, with NAFLD being the most common chronic liver disease in the United States and in other industrialized nations, highlighting a critical need to develop new therapeutic approaches [6,7].

Over the last decade, a substantial body of research has focused on the role of gut microbiota composition, microbial metabolism, and gut barrier function in the susceptibility to, development, and outcome of these conditions (reviewed in [8,9]). The liver is directly connected to the gut through the portal system and the bile ducts and, thus, is continuously exposed to gut-derived microorganisms and their metabolites. In turn, the liver can influence the composition of the gut microbiome through secreted bile acids. Therefore, if changes in the gut microbiome can cause ALD and NAFLD, modification of the composition and metabolites of the gut microbiome could be used to treat both conditions. Various strategies have been investigated, both in animal models and in human studies, that can be divided into three types of approach [10]: (1) the use of microorganisms, as in the case of fecal microbiota transfer (where the entire microbiome is transplanted), probiotics (living bacteria), fiber and prebiotics (groups of nutrients that promote the expansion of specific bacteria), symbiotics (combinations of probiotics and prebiotics), or engineered bacteria capable of producing a beneficial metabolite or of metabolizing toxic products; (2) the removal of harmful bacteria (using antibiotics or specific viruses called bacteriophages), and, finally; (3) the use of microbe-derived metabolites (also called postbiotics) and their related signaling pathways.

Among these strategies, the use of fiber and prebiotics is an attractive and relevant option, as their role in human health has been established [11] and their daily consumption has dramatically decreased with the current high prevalence of the western diet and with alcohol misuse [12]. The recommended daily intake of fiber ranges from 30 to 38 g/day for men and 21 to 25 g/day for women [13], whereas patients with NAFLD or ALD have a low daily fiber consumption, estimated to be approximately 15 to 20 g/day for men and 13 g/day for women [12]. Moreover, low fiber consumption is associated with the prevalence of NAFLD [14]. Conversely, a high-fiber and low-fat diet has been shown to be related to the regression of NAFLD [15]. The use of different fibers, such as inulin, in the context of metabolic diseases, and more particularly metabolic liver diseases, has been addressed by several studies [16,17,18]. Among the different fibers available, pectin is a soluble fiber found in different fruits and vegetables. It modulates the gut microbiome and, in recent studies, has shown promising results in protecting the liver from metabolic injuries, such as those caused by alcohol and the western diet. This review provides an overview of the biological effects of pectin application with an emphasis on its role as a microbiome-editing strategy and its potential role in modulating metabolic liver disease.

## 2. Biological Effects of Pectin

Dietary fiber consists of carbohydrates, mainly provided by fruits and vegetables, that resist digestion and absorption in the small intestine of humans. Dietary fiber is generally divided into soluble and insoluble fiber. Insoluble fiber, such as cellulose, usually found in bran, vegetables, and nuts, is generally poorly fermented by the intestinal microbiota (IM) in humans and increases the gut transit rate. Conversely, soluble fiber, such as pectin, gums, xyloglucans, inulin, maltodextrins, starch and polydextrose are highly fermentable by bacteria of the large intestine [19]. Soluble fiber is found in vegetables, whole grains, such as oats and barley, and fruits, in particular, the peels of apples and citrus, which are highly enriched in pectin (Table 1 [20,21,22,23,24,25,26,27,28,29,30,31,32,33,34]).

Pectins are complex heteropolysaccharides mainly composed of linear galacturonic acid (GalA) chains, called homogalacturonan, and complex side-chains, named rhamnogalacturonan (RG), which link to GalA [35,36,37]. The branched chains are composed of several neutral sugars, including rhamnose, fucose, and arabinose. Moreover, GalA can be both methyl-esterified and acetylated. The degree of methyl-esterification has an impact on the functional properties of pectin, which is classified as low (≤50%) or high (>50%) methoxy pectin depending on the degree of methylation [38] (Figure 1). The structure of pectin modulates nutrient absorption and gut bacteria composition and their respective production of metabolites [35].

Gut bacteria break down complex polysaccharides through the expression of a large panel of carbohydrate-active enzymes (CAZymes). Dietary fiber consumption increases the relative abundance of bacteria with CAZyme-encoding genes [39,40]. However, the changes in microbiota composition depend on the type of fiber used. Studies using in vitro microbiota systems that produce highly controlled conditions of pH and substrate supply showed that the two different soluble fibers pectin and inulin have different effects. The major microbial modifications induced by an inulin-enriched diet were an increased proportion of the *Bifidobacterium* genus and a decreased level of unclassified *Clostridiales*. Inulin also induced a specific increase in the abundance of *Bacteroides uniformis, B. caccae,* and *Anaerostipes hadrus* [41]. Compared to inulin, pectin specifically favored the growth of *Bacteroides* [38,42], with a specific increase in the abundance of *B. vulgatus/dorei, B. stercoris, B. eggerthii, B. cellulosilyticus/intestinalis, B. ovatus, B. thetaiotaomicron* and *Eubacterium eligens*. Pectin also influences the growth of the genera *Ruminococcaceae* and *Lachnospira*, including the species *Lachnospira eligens* and *Faecalibacterium prausnitzii*. It is of note that the growth of *L. eligens* is unique to pectin substrates [42].

The effect of pectin on the microbiota is, however, dependent on its chemical structure; the degree of methyl esterification, the homogalacturonan-to-rhamnogalacturonan ratio, and the molecular weight can induce specific effects on the composition of the microbiota. For example, the molecular weight particularly influences the growth of *Bifidobacterium* spp. [43]. Several in vivo studies to investigate changes in the gut microbiota induced by pectin in the context of a chow diet have been conducted. The main findings were an increase in the abundance of *Bacteroidetes* (phylum level) and *Bacteroides* (genus level) and a decrease in that of *Firmicutes* (phylum level) observed in rats and mice [38,44,45]. In vivo studies also showed specific changes in bacterial growth depending on the chemical structure, as mentioned above; sugar beet pectin, a highly methylated form of pectin with a high arabinose and galactose content (see next paragraph) significantly stimulated the growth of *Lactobacillus* and *Lachnospiraceae* [46].

By modulating the gut bacterial ecosystem, soluble fiber also induces the production of a large variety of bacterial metabolites. However, their physico-chemical properties per se modulate absorption and excretion of specific nutrients in the gut. Thus, the effects of fiber differ along the gastrointestinal tract, with an impact on nutrient absorption in the small intestine and a large impact on metabolic function/production in the colon. This dual effect of pectin is, therefore, relevant for nutritional liver diseases, such as ALD and NAFLD.

### 2.1. Physicochemical Properties of Pectin Modify Metabolite and Nutrient Availability

Pectin can form a tri-dimensional crystalline network that entraps water and small molecules. These properties, widely used in the food industry, also have an impact on the intestinal absorption of nutrients [47]. Several molecules with broad metabolic functions, such as steroids, including cholesterol and bile acids (BA), can be trapped in the gel structure formed by pectin, blocking their absorption. Therefore, pectin has been widely studied for its cholesterol-lowering effects.

Pectin has been shown to lower blood and liver cholesterol levels in various animals, including hamsters [48,49], rats [50], and obese mice [51,52,53,54,55]. However, the ability of pectin to lower cholesterol levels is mainly dependent on its molecular weight and the degree of methyl esterification [38,56]. Despite such results in animal studies, human studies focusing specifically on pectin have been scarce, but a specific diet high in apples, rich in polyphenols and pectin, showed a cholesterol-lowering effect in healthy humans [57]. As in animal models, this effect of pectin is related to its molecular weight and degree of methylation, both of which are dependent on the source of pectin. Indeed, citrus and apple pectin have been shown to be more effective than orange pulp pectin [58] in this context.

The sequestering effect of pectin is not limited to cholesterol, but is also observed for other sterols, including BA, which are excreted in feces [56,59,60,61,62,63]. BA participate in the solubilization of cholesterol in the gallbladder and promote the intestinal absorption of cholesterol, lipids, and fat-soluble vitamins. However, BA are also the ligands of receptors that induce several signaling pathways. Thus, BA in the gut control the liver synthesis of new primary BA through a negative feedback loop. They act as ligands for the farnesoid X receptor (FXR), inducing the production of fibroblast growth factor (FGF)15 in the ileum in mice and FGF19 in humans. FGF15/19 activate FGF receptor 4 in the liver, which represses the expression of the cytochrome P450 family 7 subfamily A member 1 (Cyp7a1), a key enzyme of BA synthesis. By decreasing the luminal content of BA, pectin reduces the feedback inhibition of Cyp7a1. Consequently, BA synthesis is increased, contributing to the cholesterol-lowering process [60,64]. BA also act as ligands for Takeda-G-protein-receptor-5 (TGR-5), a G-protein-coupled receptor (GPCR or GPR) that induces the production of glucagon-like peptide 1 (GLP-1) by colonic enteroendocrine cells. Using TGR5-deficient mice, or by pharmacological activation of TGR5, it has been shown that activation of TGR5 signaling attenuates hepatic triglyceride (TG) storage and fibrosis. An improvement in liver function has also been shown to be associated with dampening of the pro-inflammatory phenotype of liver macrophages (MO), including Kupffer cells [65].

Pectin can also modulate lipid absorption and its use was shown to be associated with an ε polylysine-induced reduction in serum total TG levels, in addition to cholesterol levels, and an increase in fecal excretion of TG in mice [52]. However, these lipid-lowering effects have not been systematically reproduced in animal studies [50,63]. In humans, the lipid-lowering effect was not conclusive in the few studies that have used purified pectin [58]. In contrast to the cholesterol-lowering effect that was observed in dietary interventions, including those enriched in apples, the plasma lipid content was not modified [66]. A recent review provided a comprehensive list of plant compounds and their potential lipid-lowering effects, but whether these compounds, including pectin, have a true lipid-lowering effect still needs to be demonstrated [67].

The pectic gel that forms in the small intestine has an impact not only on lipid and sterol absorption, but also on that of glucose. Several reports addressed this point in the 1980s and 1990s and showed that pectin (5%) decreases jejunal glucose absorption and improves glucose tolerance in rats and mice [50,68]. Moreover, in mice, one month pectin treatment was found to reduce fasting glucose levels [44]. In humans, an apple-enriched diet also improves glucose homeostasis [69]. In healthy humans, it has been shown that pectin (10 or 15 g of pectin per day) given before a glucose challenge can impair the intestinal absorption of glucose and, thus, help to decrease postprandial glycaemia [70].

The effect of pectin on bodyweight gain and food intake has been described using rodent models and may depend on the formation of pectin gels in the gut. However, this effect was observed for diets containing up to 10% pectin and was not specific to pectin, but rather to a general soluble fiber effect, as other sources of soluble dietary fiber produced significant effects on bodyweight and food intake [71,72,73]. Despite such results, no clear dietary strategies have emerged from these studies [74].

The role of pectin in alcohol absorption has recently been addressed in animal models of ALD. In such models of chronic alcohol administration, plasma ethanol did not differ between alcohol-fed mice treated or not with pectin [75].

### 2.2. The Fermentation of Pectin by Gut Bacteria Produces Active Metabolites

Among a large panel of metabolites that can be produced by gut bacteria, short-chain fatty acids (SCFA) are involved in the molecular mechanisms mediated by pectin. They exhibit pleiotropic effects on lipid and glucose homeostasis that can be complementary or opposite [76]. SCFA are principally transported to the peripheral circulation via the portal vein and can act on the liver and peripheral tissues. They can serve as an energy source for colonocytes or as signaling molecules through GPCRs [76]. Bacteria that ferment fibers, including pectin, produce mainly acetate, propionate, and butyrate [77]. As the composition of the microbiome is dependent on the chemical structure of pectin, the production of SCFA is also modulated by the chemical features of the type of pectin. Thus, low methyl-esterified citrus pectin and complex soy pectin have been shown to increase the production of total SCFA, propionate and butyrate, whereas high methyl-esterified pectin and sugar beet pectin do not [35].

In the intestine, SCFA can be used locally by gut bacteria, with butyrate mainly serving as an energy source for colonocytes. As GPR41 (also known as FFAR3 - free fatty acid receptor 3) and GPR43 (FFAR2) ligands, SCFA promote the expression of peptide YY (PYY) and GLP-1 by enteroendocrine cells. Both these peptides slow down the intestinal transit and decrease food intake. PYY mainly induces anorexigenic signaling and GLP-1 controls post-prandial glycemia through an increase in insulin secretion and inhibition of glucagon secretion [78,79]. It has also been shown that pectin stimulates intestinal mucus secretion in rodents through the activation of immune cells by SCFA, thus participating in the maintenance of gut epithelial integrity (see the paragraph below) [75,80,81,82].

In the liver, propionate serves as a precursor for de novo gluconeogenesis and inhibits fatty acid synthase expression and lipogenesis. Conversely, acetate and butyrate may be involved in lipogenesis. Through the activation of GPR41/GPR43, acetate and butyrate can activate AMPK (AMP-activated protein kinase) phosphorylation and peroxisome proliferator-activated receptor α (PPARα) target genes, both favoring fatty acid oxidation (FAO) and glycogen storage [78].

In the pancreas, acetate, butyrate, and propionate act through GPR43 to modulate the glucose-stimulated insulin secretion, and GPR41 contributes to the regulation of β cell mass [83]. In muscle, acetate and butyrate activate GPR41/GPR43, inducing PPARδ expression and subsequent FAO and AMPK phosphorylation, inducing glucose storage. In white adipose tissue (WAT), propionate, butyrate, and acetate activate PPARγ expression and, thus, adipogenesis. Propionate can increase free fatty acid uptake and, with acetate and butyrate, decrease MO infiltration and inflammatory cytokine/chemokine levels.

SCFA are also involved in the modulation of intestinal immune homeostasis [84], as they are able to modulate the activation of MO, dendritic cells (DC), innate lymphoid cells (ILCs), and T-cells through the binding of GPCRs, including GPR41, GPR43 and GPCR109A (also known as hydroxycarboxylic acid receptor, HCAR2) [85,86]. Some of these effects depend on SCFA-mediated regulation of histone acetyltransferase (HAT) and histone deacetylase (HDAC), which orchestrate the post-translational modifications of transcription factors. Thus, SCFA can decrease the production of inflammatory cytokines by neutrophils, MO, and DC. In DC, SCFA promote a tolerant profile. Butyrate also promotes regulatory T-cell expansion in the colon by inhibiting HDAC activity. The responses of ILCs, including proliferation and activation, and their role in the production of mucus and antimicrobial peptides, are also shaped, in part, by SCFA [87].

Butyrate, acetate, and propionate are involved in the storage of hepatic lipids either by activating the GPR41/GPR43 receptors in the liver or those located within other target tissues, including the intestine, muscle, and white and brown adipose tissues [79]. Moreover, they influence glucose homeostasis through GLP-1 production and the modulation of gluconeogenesis and glycogen storage. These effects all depend on the production of each SCFA and their respective affinity for GPCRs. GPR 41 binds to propionate with a high affinity, then to butyrate, and, finally, to acetate. GPR43 binds with a similar affinity to propionate and acetate and GPR109A mainly binds to butyrate [76]. Furthermore, metabolites derived from protein or other dietary nutrients also interact with a large panel of GPCRs to regulate these metabolic pathways [88].

Such complex regulation could explain why clinical trials using fibers, which have shown increased SCFA levels depending on the physicochemical properties of fibers, have failed to demonstrate any clear beneficial effect on glucose homeostasis in humans.

## 3. Pectin Alleviates NAFLD/MAFLD

NAFLD, including NASH, are strongly linked to overweight and obesity, type II diabetes mellitus, and the metabolic syndrome [3]. This large overlap between NAFLD and metabolic disorders makes it difficult to dismantle the specific effects of dietary interventions on liver injury from their effects on the metabolic syndrome and its specific features. Indeed, a western-style diet, enriched in fats and sugars and low in fibres, induces prolonged metabolic stress that leads to adipose tissue dysfunction, inflammation, and the release of adipokines that trigger liver injury, as well as type II diabetes and dyslipidemia.

The role of pectin in its effect on liver injury in rodent models has previously been investigated. The diversity of the diets used to induce NAFLD/NASH and of the types of pectin used make it difficult to compare the studies involved. We focused on publications that evaluated the impact of pectin on liver injury and summarize the metabolic and microbiota changes, when examined, induced by pectin (Table 2). The amount of fiber and single sugars contained in the carbohydrate content is specified, if reported in the publication. Several studies have assessed the improvement in liver function using a chow diet enriched with pectin in lean or obese rodent models. In C57BL/KsJ db/db mice, a rodent model of obesity and insulin resistance, one month of pectin treatment specifically reduced liver steatosis [89]. Pectin also improved hyperglycemia in association with an improvement in hepatic glycogen metabolism. This improvement in glucose homeostasis was mediated by activation of the insulin and AMPK signaling pathways, including activation of the signaling cascades of insulin receptor substrate-1 (IRS-1) and AMPK [89]. In addition to the lower expression of glucokinase, phosphoenolpyruvate carboxykinase, and glucose-6-phosphatase, pectin treatment increased glycogenesis and decreased glycogenolysis and gluconeogenesis [89]. In lean elderly Wistar rats (seven months old), one month of pectin treatment improved liver and WAT insulin and leptin resistance associated with a decrease in plasma leptin level. Of note, in this study, the pectin-treated rats were compared to a pair-fed caloric-restricted group to discriminate the effects of pectin from those related to lower caloric intake, which is generally associated with dietary fiber. The pectin-treated rats had a lower body-fat content and a decreased homeostatic model assessment for the insulin resistance (HOMA-IR) index than the pair-fed rats. From a mechanistic point of view, the pectin-treated rats showed decreased expression of genes related to energy uptake and lipogenesis in the WAT. Conversely, they showed decreased expression of genes related to lipogenesis and increased expression of those involved in lipolysis and FAO in the liver (Table 2), which could be partially attributed to the concomitant reduction in caloric intake [71].

In studies using a cholesterol-enriched diet, the effect of pectin on liver homeostasis was shown to be related to cholesterol metabolism [90,91]. As mentioned above, both pectin and guar gum induced a significant cholesterol-lowering response associated with upregulation of hydroxy-methyl-glutaryl coenzyme A (HMG-CoA) reductase and cholesterol 7α-hydroxylase. A decrease in acyl CoA cholesterol acyltransferase (ACAT) and low-density lipoprotein (LDL)-receptor levels was observed only in pectin-treated guinea pigs [90]. In a more recent publication, the regulation of the BA enterohepatic cycle was examined through the FXR-FGF15 signaling pathway. The authors showed that the pectin-induced BA fecal excretion was associated with a decrease in the level of FXR in the small intestine of mice and subsequent lower FGF15 and higher hepatic expression of Cyp7a1. In this study, apical sodium-dependent BA transporter (ASBT) expression in the small intestine was increased in pectin-treated mice, suggesting that this BA transporter does not compensate for the sequestering effects of pectin [60].

Using a mixed fat and cholesterol-enriched diet, pectin treatment, for at least four weeks and up to 14 weeks, induced a reduction in bodyweight gain and serum TG levels and increased fecal lipid excretion [92,93]. Pectin improved lipid homeostasis in the liver by reducing steatosis and modulating lipid metabolism, including increasing the FAO-related enzyme activities of acyl-CoA oxidase and carnitine palmitoyl transferase 1 (CPT1), as well as upregulating PPARα [92,93]. Moreover, pectin improved hepatic antioxidant capacity by increasing antioxidant enzyme activity (i.e., superoxide dismutase, catalase, and glutathione peroxidase) [92].

More recently, several studies used a high-fat diet enriched in fiber to treat liver injury during metabolic syndrome. Pectin was able to decrease bodyweight gain, with a minimum dose of 8%, as previously described [71,94,95,96,97,98,99,100]. This decrease in bodyweight was associated with a decrease in fat mass and an increase in the level of transcription factors involved in the CCAAT/enhancer binding protein (C/EBPα)/PPARγ pathways in WAT [101]. Browning of WAT was also described, without a clear functional study, concerning adipocyte lipolysis [98,102]. Moreover, two studies reported a decrease in plasma leptin levels in pectin-treated groups [95,99] and an increase in PYY [95], which could at least partially explain the effects on weight gain (Figure 2).

As already described, cholesterol-enriched diets often improve plasma cholesterol [94,97,98,103], fasting blood glucose [96,100], and insulin [72,99] levels. However, an improvement in transaminase levels was observed in only two studies [97,98], in contrast to an improvement in steatosis, which was associated with a decrease in hepatic TG [94,95,96,98,102,103] and/or cholesterol levels [94,96,98]. This was associated with an improvement in hepatic lipid metabolism, including a decrease in the expression of genes involved in lipogenesis [98,101,103] and the AMPK signaling pathway [101]. In addition, pectin has been shown to change hepatic lipid content by reducing saturated fatty acid (SFA) and mono-unsaturated fatty acid (MUPA) levels and increasing poly-unsaturated fatty acid (PUFA) levels [98,103]. This could be due to the fact that pectin, even in small amounts (4% and 8%) in the diet, reduces hepatic lipid peroxidation and oxidative stress [97,98].

As described with respect to a chow diet, pectin added to a high-fat diet (HFD) increased the abundance of *Bacteroides* [94,98,99,102,104], except in one study that used 8% pectin [96]. Although the effect on the relative abundance of *Firmicutes* is not clear, as a number of studies have reported an increase [98], while others have reported a decrease [102], the *Bacteroides*/*Firmicutes* ratio has been shown to consistently increase. An increase in the abundance of mucin-degrading *Akkermansia* was observed in two studies [94,96]. This is a relevant modification induced by pectin, as the abundance of *Akkermansia* has been reported to be decreased in NAFLD patients [8] and, conversely, the administration of *Akkermansia* decreased liver steatosis in animal models of NAFLD and ALD [105,106]. A decrease in the abundance of *Proteobacteria* was also reported in two studies [99,102]. Data concerning other differences observed in the relative abundance of several members of the IM are conflicting.

In several studies, SCFA content was determined in association with changes in the composition of the gut microbiota. Serum and/or cecal SCFA content increased [94,96,98], with a specifically higher level of acetate in serum and the caecum [94,96,98,102,104]. Serum propionate was also increased [94], as well cecal propionate [98]. In the caecum, a diet with 8% pectin resulted in a decrease in valerate, isovalerate and isobutyrate levels [98]. However, these studies did not report any information concerning the effect of these SCFA through their GPCR receptors.

## 4. Pectin Improves Alcoholic Liver Disease

Alcohol-induced liver disease is associated with changes in the composition and function of the IM in both humans and mice [107,108,109,110]. Among these changes, a common finding is a decrease in the abundance of *Bacteroides*. Moreover, a microbiome with a low level of *Bacteroides* can increase the susceptibility of mice to developing ALD [75,81]. As pectin can induce an increase in the abundance of *Bacteroides*, the use of pectin in a mouse model of ALD alleviated steatosis and liver inflammation and improved leaky gut (Table 3). The use of pectin in this model was dose-dependent and induced major changes in the microbiome composition and function. Pectin induced an increase in the abundance of *Bacteroides* and changes in the fecal metabolome [75]. Among the pathways that were altered, the authors showed that pectin treatment induced an increase in the level of indole derivatives, which are bacterial tryptophan metabolites and potent agonists of the aryl hydrocarbon receptor (AhR). Treatment with a synthetic AhR agonist in the murine ALD model induced similar effects on liver steatosis and inflammation, and on the gut barrier, through an increase in IL22 level and enhancement of mucus and anti-microbial peptide production. However, pectin still decreased liver inflammation in AhR-deficient mice fed alcohol, suggesting that its effects are not solely mediated by bacterial tryptophan metabolites. Indeed, among the other metabolites that can be modulated by pectin and that could play a crucial role in ALD, BA could act synergistically with tryptophan metabolites (Figure 3).

Pectin restores the enterohepatic BA cycle following alcohol-induced dysregulation and leads to a decrease in plasma and hepatic BA levels, an increase in caecal BA levels, and changes in the overall composition of the BA pool, which shifts towards being more hydrophilic, including an increase in free or tauro-conjugated ursodeoxycholic acid (UDCA and TUDCA), which are less hepatotoxic [111]. This is due to an increase in the abundance of bacteria capable of processing and metabolizing BA, notably *Bacteroides* and *Enterobacteriacae*. However, the effects of pectin on the enterohepatic BA cycle are indirect due to its sequestering properties, rather than by directly modulating the FXR/FGF-15/19 pathway. Nonetheless, the administration of pectin to alcohol-fed mice leads to modifications in BA signaling in several organs, including the gut, liver, and brown adipose tissue, thus alleviating ALD [111].

Pectin could also be used in symbiotic combinations in ALD. A recent study showed that the administration of pectin with *B. fragilis* ATCC25285 resulted in a better protective effect against ALD than the individual agents used alone [112]. In this context, pectin improved *Bacteroides* colonization and modulated the metabolic capacity of the microbiome, leading to an increase in SCFA (acetic acid, propionic acid, and butyric acid) and the production of more tryptophan metabolites that are AhR agonists (indoleacetic acid, indole-3-propionic acid, indoleacetic acid) [112].

Although pectin has shown promising results in animal models of ALD, there are currently no data on its use in this context in humans.

## 5. Effect of Pectin on Hepatocellular Carcinoma (or Cancer)

In humans, epidemiological studies that have analyzed the role of dietary fiber in the risk of developing various types of cancer have mainly been based on daily fiber intake included in the meal, with no conclusive results. However, these studies generally suggest that a high-fiber diet is associated with a lower risk of developing colorectal cancer [113]. Moreover, as the gut microbiota has been shown to be involved in the efficacy of the response to immune checkpoint inhibitors [114,115], one study investigated the role of pectin in tumor-bearing mice. A low dose of pectin (10 mg/kg per day), which was associated with an increase in butyrate levels, improved the response to an anti-programmed-cell-death protein 1 (PD-1) immune checkpoint inhibitor in colorectal tumor-bearing mice [116]. In mice that develop liver metastases in a colon cancer model, the administration of pectin for three weeks reduced the levels of galectin-3, an oncogenic protein that regulates cell homeostasis, including growth and adhesion [117]. Rhamnogalacturonan II (RG-II, see Figure 1), a component of pectin that can be produced through bacterial fermentation, showed a preventive effect against lymphoma by increasing the DC-based immune response through the toll-like receptor 4 (TLR4) signaling pathway. However, this indirect beneficial effect of pectin in cancer has not been tested in liver cancer [118].

In a study using a chemically induced model of hepatocellular carcinoma (HCC) (2,6-dinitrotoluene), a pectin-containing diet protected rats from the development of HCC [119]. However, the role of pectin as a steroid-sequestering molecule could have mediated the main protective effect in this study. In a more recent publication, a possible adverse effect of pectin was highlighted [120]. In this study, mice deficient for the receptor TLR5 were used. The authors described the deleterious effect of an inulin-enriched diet in the development of liver tumors associated with high plasma bilirubin levels. Antibiotics decreased the number of liver tumors and, conversely, wild-type littermates co-housed with TLR5-deficient mice developed liver tumors. Both experiments demonstrated the involvement of the gut microbiota in development of the cancer. To a lesser extent, pectin also induced liver tumors. However, the use of cholestyramine, a BA-sequestering molecule, dampened tumor development, suggesting that the similar sequestering properties of pectin could be involved in the weaker tumor development compared to inulin. Of note, TLR5-deficient mice are not a common model of HCC. However, these results highlight a possible side-effect of fiber, in general, and of pectin, in particular, that requires further investigation.

## 6. Conclusions and Perspectives

As described in this review, the use of pectin as a soluble fiber, in the specific management of the liver diseases, ALD and NAFLD, has shown promising results, but has been mainly documented in rodent models. A daily dose of pectin of approximately 10% in the diet can improve liver damage in NAFLD and ALD through changes in the gut microbiota and the production of its metabolites, including SCFA, BA and bacterial indoles. However, pectin also acts through its physicochemical properties by forming a pectic gel in the gut, which is suspected to contribute to the decrease in food intake involved in the reduction of weight gain in NAFLD/HFD rodent models. In studies in which animals have a lower food intake, it may be difficult to know whether the improvement in liver injury is due to the decrease in bodyweight associated with improved glucose homeostasis or to a direct effect of pectin on metabolism. This point needs to be better deciphered in further studies using pair-fed animals, as described in one study [71].

The acceptability of high amounts of pectin is reduced by its poor palatability and side-effects, which include increased abdominal discomfort and intestinal pain, which compromise its use by some individuals. These effects are difficult to address in rodent models. Nevertheless, dietary interventions in humans need to be personalized to find a treatment with the lowest digestive side-effects and the most efficient modifications of IM function for each individual. Furthermore, it is important to distinguish the respective beneficial effects of pectin between its physicochemical properties and its impact on the gut microbiota to decrease the amount of required pectin and develop treatments based on ‘beneficial’ bacteria or/and their metabolites.

An alternative solution is the use of symbiotics (combinations of pre- and probiotics) and/or postbiotics (bacterial small molecules). One study showed that the use of *Bacteroides fragilis* in combination with its substrate, pectin, improved the effect on ALD [112]. This could make it possible to decrease the dose of pectin administered and improve tolerance. In addition, in other contexts, the use of higher-fermented foods increased IM diversity and decreased inflammation compared to a high-fiber diet [40]. The postbiotic alternative was shown to improve ALD using indole derivatives in a rodent model [121]. Nevertheless, more studies are needed to investigate these potential combinations and their effect in these conditions. Finally, as the changes in microbiota composition observed after prebiotic and probiotic interventions depend on the initial composition of the IM [122,123], personalized nutritional intervention may be needed to better target the type of diet that would provide the most benefit to patients.

As described above, the use of pectin in both NAFLD and MAF has shown promising results in animal models of these conditions, but no clinical trials in humans have yet been published. Four clinical trials (one recruiting and three completed) concerning patients who are overweight are registered on clinicaltrials.gov but focus more on cardiac and metabolic outcomes and not on liver-related endpoints. Moreover, their results are not yet available. Therefore, clinical trials are needed to confirm the effects of pectin on both NAFLD/NASH and ALD in humans.

## Figures and Tables

**Figure 1 nutrients-15-00157-f001:**
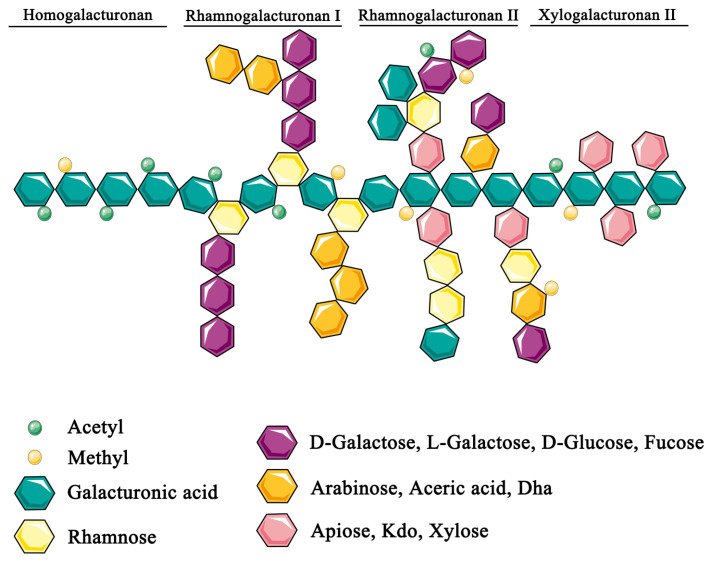
Chemical structure of pectin composed of linear galacturonic acid (GalA) chain and complex side-chains including homogalacturonan, rhamnogalacturonan I (RGI), rhamnogalacturonan I (RGII) and xylogalacturonan (XG). Abbreviations: Dha, 3-deoxy-D-lyxo-2-heptulosaric acid; Kdo, ketodeoxyoctonic acid.

**Figure 2 nutrients-15-00157-f002:**
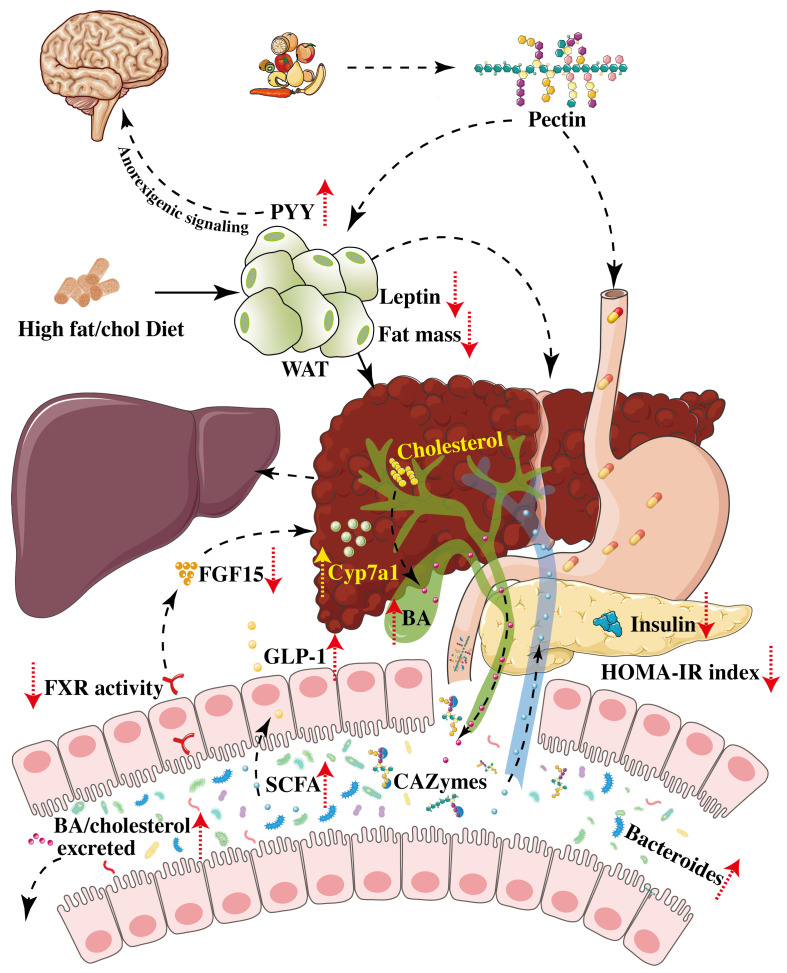
Graphical summary of mechanisms involved in pectin-induced changes in non-alcoholic fatty liver disease (NAFLD). High-fat and high-cholesterol diets lead to disorders of adipose tissue metabolism, inflammation, etc., thereby participating in liver damage. Complex polysaccharides of pectin are broken-down by CAZymes expressed by gut bacteria to make pectin work. Glucose homeostasis was improved by decreasing the plasma insulin concentration and HOMA-IR index and improving WAT insulin and leptin resistance associated with a decrease in plasma leptin and an increase in plasma PYY that induces anorexigenic signaling to the brain. Pectin can also decrease fat mass in WAT. Pectin increases Bacteroides and SCFA, but decreases FXR level in the intestine, induces a subsequent lower FGF15, and increases hepatic expression of Cyp7a1, which promotes the conversion of cholesterol to BA. The dotted line represents the pectin treatment process; the solid line represents the liver injury process. Abbreviations: BA, bile acid; CAZymes, carbohydrate active enzymes; Cyp7a1, cholesterol 7α-hydroxylase 1; FGF15, fibroblast growth factor 15; FXR, farnesoid X receptor; GLP-1, glucagon-like peptide 1; HOMA-IR index, homeostatic model assessment for insulin resistance index; PYY, peptide YY; SCFA, short-chain fatty acids; WAT, white adipose tissue.

**Figure 3 nutrients-15-00157-f003:**
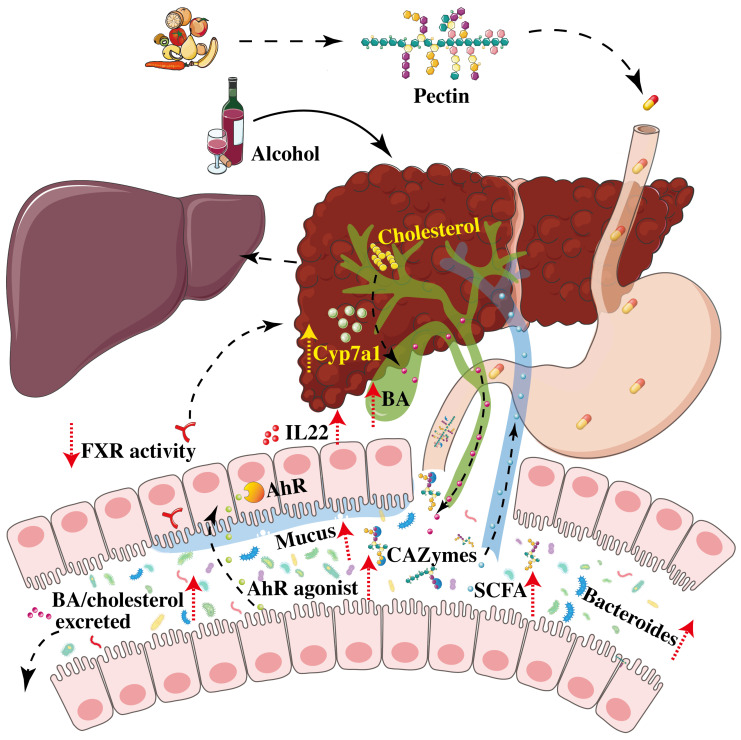
Graphical summary of mechanisms involved in changes induced by pectin in alcoholic liver disease (ALD). Complex polysaccharides of pectin are broken-down by CAZymes expressed by gut bacteria to make pectin work. Pectin increased in Bacteroides, SCFA, and indole derivatives, such as AhR agonists. Pectin increases liver BA synthesis by decreasing FXR activity and increasing the level of Cyp7a1, which promotes the conversion of cholesterol to BA. The dotted line represents the pectin treatment process and the solid line represents the liver injury process. Abbreviations: AhR, aryl hydrocarbon receptor; BA, bile acid; CAZymes, carbohydrate active enzymes; Cyp7a1, cholesterol 7α-hydroxylase 1; FXR, farnesoid X receptor; SCFA, short-chain fatty acids.

**Table 1 nutrients-15-00157-t001:** Sources and physicochemical properties of pectin.

Pectin Sources	Yield (%)	DM (%)	Mw (kg/mol)	Reference
Apple pomace	4.2–25.3	41.7–96.02	142–899	[20,22,23,24]
Banana peel	2–9	40–80	87–248	[22,28]
Beet pulp	20.0–24.87	52–58.92	116–311	[20,32]
Carrot pomace	5–15.2	45.2–77	114–1460	[20]
Chicory	12.2	44.7	260	[20]
Citron peels	13.4–37.52	37.5 -82.2	342.7 -918	[20,32]
Cocoa pod husks	4.2	8.1	-	[20]
Cubiu fruit	14.2	62	628	[20]
Eggplant peel waste	26.1	60.2	-	[20]
Fresh watermelon rinds	19.3	63.0	34.51	[20]
Gardenia jasminoides J. Ellis flower	18.04 ± 1.81	32.76 ± 1.58	141.50 ± 52.09	[21]
Grape pomace	3.96–11.23	62.14–83.11	41.5–53	[30,31]
Grapefruit peel	25–30	67.59, 69.03	132.01, 385.5	[20,28]
Green tea leaf	5.3–9.2	21.1–26.5	276–396	[20]
Jackfruit rinds	14.59	-	-	[20]
Lime peel	13–26	78.49	794.7	[20,28]
Lycium ruthenicum	3.1–7.31	2.96–31.03	38.24–5291	[34]
Lyophilized watermelon rinds	14.2	61.5	40.39	[20]
Mango peel residues	1.36–20.9	70–88.38	14.13–2858	[22,26,29,32,33]
Medlar fruit	-	62.9	198	[20]
Orange peel	24	37	-	[28]
Papaya peel	16	53.4	-	[20]
Passion fruit	10–14.8	9.57–60	802	[20,28]
Pomegranate peel	8.5	75	549	[20]
Pomelo peels	6–37	57.87	353	[20,28]
Ponkan peel	25.6	85.7	86.0	[20]
Potato pulp	14.34	37.45	320	[20]
Pumpkin waste	7.4	3–18	139–289	[22,28]
Sesame seed hull	0.03–8.07	33.11–41.53	22.7–44.6	[25]
Stems of *E. arvense*	5.9	16	360	[20]
Sugar beet pulp	7.1, 24	28–52	651, -	[20,28]
Sweet prickly pear	-	26.83	204.08	[20]
Tomato	7.55–32.6	45.7–88.98	19	[24]
Unripe banana	11.63	-	-	[20]
Watermelon peel	2.1–28	41.2–87.28	34.9–119	[20,27,28]

Abbreviations: DM, degree of methyl esterification; Mw, molecular weight.

**Table 2 nutrients-15-00157-t002:** Changes induced by a pectin-enriched diet to address liver injury in high-fat rodent models and non-alcoholic fatty liver disease. Summary of metabolic changes induced by pectin in studies addressing the improvement of liver injury in non-alcoholic liver injury.

	Animal Species	Animal Model	Type of Pectin	Pectin Amount % or g/day/kg	Duration in Days and (weeks)	Weight Gain Fat Mass Adipose Tissue	Liver Steatosis Liver Lipid Metabolism	Plasma Lipids Plasma Metabolites	ALT Liver Inflammation Liver Metabolism	Bile Acid and Cholesterol Metabolism or Metabolites SCFA	Glucose Homeostatis	IM Composition and Gut Homeostasis	Ref
**Chow Diet**	C57BL/Ksj db/db mice, male	Standard chow diet UNK	Ficus pumila Linn Pectin HM	100 or 200 mg/kg/day gavage	curative 28 days (4 w) in 12 w	no effect on BW, Food & water intake	↓ steatosis **mRNA & protein** ↓ G6Pase, PEPCK, pIRS-1, pGS ↑ GK, pAkt, pGSK3β & pAMPKα				↓ fasting blood glucose ↓ serum insulin ↓ HOMA-IR ↑ liver glycogen		[89]
	Wistar rats, male	Standard chow diet UNK 3.3 kcal/g 8% kcal from fat 4% cellulose	Apple pectin HE	10%	preventive 30 days (1 m)	↓ BW gain and cumulative food intake ↓ fat content ↑ lean mass **WAT:** ↓ Prkaa2, IRS1, AKT, pAKT, Pparγ, Acaca, Fasn, Gpam, Scd1, Lpl, Slc2a4, Pnpla2 ↑ STAT3, pSTAT3, AMPK	↓ Irs1, pIrs1, Prkaa2, Lepr, AMPK & pAMPK, Srebf1, Mlxipl, Gpam, Fasn, Scd1, ACC ↑ STAT3 & pSTAT3, Pnpla2, CPT1,	↓ plasma leptin ↑ adiponectin = ↓ L/A ratio			↓ fasted blood glucose and insulin ↓ HOMA-IR index		[71]
**High Chol Diet**	Hartley guinea pigs, male and female	Protein 23% Fat 15.1% Carbohydrate 52.1% Simple sugar UNK Fiber 12.5% Chol 0.04%	Lime peels pectin vs. cellulose as control vs. gum guar	12.5%	preventive 28 days (4 w)		↓ free & total & esterified Chol	↓ TC, VLDL, LDL	↑ HMG-CoA reductase, chol 7a-hydroxylase, apoB/E receptor ↓ hepatic ACAT ↑ reductase activity				[91]
	Hartley guinea pigs, male						↓ free & esterified Chol	↓ Chol ↓ ApoB	↑ chol 7a-hydroxylase	↓ Intestinal Chol absorption ↑ LDL FCR			[90]
	Kunming mice, males	Protein UNK Fat UNK (Lard 5%) Carbohydrate UNK Simple sugar UNK Fiber UNK Chol 2% vs. 0.4%	pectin HPPS	300 mg/kg BW oral infusion	preventive 8 days (4 w)	↓ BW gain, serum & hepatic TC level			↓ hepatic TC level	↓ BA in liver, ileal, small intestine levels, total BA pool size ↑ BA content in gallbladder, feces **mRNA or protein:** ↓ ileal FXR, FGF15, SHP1, ↑ ASBT ↓ liver FGFR4, Cyp7a1 ↑ liver Cyp7a1			[51]
**High Chol Diet**	Kunming mice, male	Protein UNK Fat 10% (Lard 10%) Carbohydrate UNK Simple UNK Fiber UNK Chol 2%	pectin HPPS GA 98%	50, 150 or 300 mg/kg BW oral infusion	preventive 28 or 70 days (4 w or 10 w)	↓ BW gain in mice fed a HFD ↓ fat accumulation	↑ FA oxidation-e activities ↑ CPT-I & 3KCT (4, 10 w), DCR & ACO (10 w), ↑ activities of peroxisomal 3KCT, ACO DCR, mitochondrial CPT-I ↑ PPARα	↓ FFA ↓ TG				↑ fecal total lipids	[93]
					preventive 70 days (10 w)	↓ BW gain ↓ eWAT ↓ perirenal fat pads in HFD fed mice	↓ liver TG, GPAT & PAP activity ↓ lipid steatosis		↑ antioxidant enzyme activities: SOD, GSH-Px, CAT, GSH & TAC ↓ MDA				[92]
**High Chol Diet**	Wistar rats, male	Protein 20% Fat 28% (Lard 23%) Carbohydrate 44% Simple sugar 10% Fiber 34% Chol 2%	Apple pectin HE 70–75% +/− guar gum	8%	preventive 14, 28, 42 days (2 w, 4 w or 6 w)	↓ BW gain ↓ fat content	↓ liver weight ↓ TG ↓ Chol	↓ Chol **6 weeks:** ↓ serum MCP-1		2 weeks: ↑ serum & cecal acetate ↑ serum propioniate ↑ serum & cecal total SCFA		↑ Bacteroides (guar gum), ↑ Akkermansia (fibre-free), great individual variance (pectin) **2 weeks:** ↑ weight of cecal content	[94]
		Protein 12% Fat 10% Carbohydrate 62.1% Simple sugar 10% Fiber 52.1 %	Citrus pectin vs. guar gum vs. FOS	10%	preventive 12 days (1.7 w)					↑ acetate in cecum and portal serum: correlation between cecum-formed and absorbed SCFA		↓ caecum tissue weight	[104]
		Protein 20% Fat 28% (Lard 23%) Carbohydrate 35% Simple sugar 10% Fiber 25% Chol 2%	Citrus peel pectin LM (24%) or HM (70%)	8%	preventive 21 days (3 w)	↓ BW gain, epididymal fat pad, liver & spleen weight, ↓ liver fat	↓ TG ↓ chol (HMp),	↓ TG (LMp), no change of chol		↑ serum & cecal SCFA ↑ acetate (HMp)—no changes for propioniate butyrate	↓ blood glucose	↑ Akkermansiano effect in Lactobacillus, Bacteroides et Bifidobacterium	[96]
		UNK except Fat 10% sheep fat	Apple pectin	0,5 mg/kg/day (gastric gavage)	preventive 49 days (7 w)	↓ BW gain ↓ eWAT ↓ perirenal fat pads in HFD fed mice		↓ serum TC, LDL-C, TG levels ↑ HDL-C ↓ TBARS level ↑ SOD, CAT and GSH-Px activities	restore normal AST, ALT ↑ SOD, CAT and GSH-Px, activities (liver, kidney) ↓ TBARS level (liver, kidney)				[97]
	Sprague Dawley rats, male	AIN-93 modified Protein 26.7% Fat 23.7% (Lard 19.4%) Carbohydrate 32.8% Simple sugar 10.5% Fiber 22.8%	Apple pectin HM and HE > 50%	10%	curative 28 days (4 w) in 11 w	↓ Final BW & BW gain ↓ fat mass & body fat percentage ↓ total lean mass ↑ total body lean	↓ liver total fat ↓ TG levels	↓ total chol & TG ↑ Plasma PYY ↓ Plasma leptin			↓ serum insulin	↓ Cumulative caloric intake ↑ small intestine and caecum weights and lengths	[95]
	Sprague-Dawley rats female	Protein 19.5% Fat 23% (Lard 21%) Carbohydrate 51% Simple sugar 34% Fiber 11%	Apple pectin (Apple pomace)	10%	preventive 56 days (8 w)		↓ fat vacuoles & histology scores ↑ palmitic acid (16:0) ↓ palmitoleic acid & oleic acid content ↓ liver DGAT2 mRNA	↓ total BA concentration					[103]
	C57BL/6J mice, male	Protein UNK Fat 30% (Lard 30%) Carbohydrate UNK Simple sugar UNK Fiber UNK Chol UNK	Citrus peel pectin GA > 74%	4% & 8%	preventive 84 days (12 w)	↓ BW gain ↓ BMI ↓ eWAT weight ↓ fat index ↓ adipocyte size	↓ TG, TC, NEFA ↓ FAS, ACC & ChREBP levels ↓ SFA, MUPA, palmitic acid levels ↑ PUFA ↓ hepatic fat accumulation	↓ TG, TC, LDL-C ↑ HDL-C	↓ ALT, AST ↓ liver NF-κB, TNFα, PPARα & MDA, p-ERK, p-JNK, p-p38, Nrf2 levels, ratios of pERK/ERK and pJNK/JNK ↑ GSH-Px, SOD activities	↓ cecal isobutyric acid, isovaleric acid & valeric acid levels ↑ cecal total SCFA, acetate & propioniate levels		↑ Firmicutes, Bacteroides, Parabacteroides, Allobaculum, Bifidobacterium, Olsenella, Barnesiella, Anaerobacterium, Clostridium IV ↓ Lachnospiraceae, Lactobacillaceae, Lactobacillus, Helicobacter, Alistipes, Clostridium XIVa	[98]
		Protein 30% Fat 40% (Lard UNK) Carbohydrate 30% Simple sugar UNK Fiber UNK	Pectin UNK	10%	curative 35 days (5w) in 17w	↓ BW (LFP diet) ↓ BW gain (HFP diet)	↓ liver adiposity				↓ fasted blood glucose		[100]
		Protein 23% Fat 23.5% (Lard 21%) Carbohydrate 46.5% Simple sugar 20% Fiber 14.3%	Apple pectin	10%	preventive 56 days (8w)	↓ BW gain ↓ fat mass	↓ liver lipid	↓ plasma leptin, resistin			↓ insulin fed (or fasted, unclear)	↑ Bacteroidetes, Proteobacteria, Deltaproteobacteria ↓ cecal Claudin5, Trefoil Factor3 gene expressions	[99]
		Protein 26% Fat 35% (Lard 31%) Carbohydrate 32% Simple sugar 9.5% Fiber 6.5%	Apple pectin	2% (0.06 g pectin/30 g of mouse = 2 g/kg)	Curative 56 days (8 w) in (16 w)	**eWAT** ↓ semi-quantified adipocyte diameter	↓ TG, liver/body ratio ↓ lipid droplet size in BAT					↓ Firmicutes, Ruminococcus, Desulfovibrionaceae, proteobacteria ↑ Bacteroidetes, S24_7, Prevotellaceae, Turicibacteraceae	[102]
	Kunming mice, male	UNK HFK Bioscience Chow and High fat diets	Hawthorn pectin oligosaccha ride (POS)	0.25, 0.75, 1.5 g/kg diet (0.025%, 0.075%, 0.15%)	preventive 70 days (10 w)	**WAT mRNA and protein** ↑ cAMP, AC, C/EBPα, PPARγ, RXR, PKA, Pap1, pSRC, pERK, pCREB	**mRNA or protein:**↑ ADPN, LKB1, ACO, CPT-1, adipoR1 (1.5 g/kg), PPARα, PGC-1α, NRF-1 (0.75 & 1.5 g/kg). For all diets ↑ AMPKα, p-AMPKα, adipoR1 ↓ ACC	↓ TG, TC, total lipids, ADPN level					[101]

Abbreviations: 3KCT, 3-ketoacyl-CoA thiolase; AC, adenylate cyclase; Acaca, acetyl-CoA carboxylase alpha gene; ACAT, acyl CoAcholesterol acyltransferase; ACC, acetyl-CoA
carboxylase; ACO, acyl-CoA oxidase; AdipoR, ADPN receptor; ADPN, adiponectin; ALT, alanine aminotransferase; AMPK, adenosine 5′-monophosphate (AMP)-activated protein kinase; ApoB, apolipoprotein B; ApoB/E, apolipoprotein B/apolipoprotein E; ASBT, apical sodium-dependent bile acid transporter; AST, aspartate aminotransferase; BA, bile acid; BAT, brown adipose tissue; BMI, body mass index; BW, bodyweight; C/EBP*α*, CCAAT/enhancer binding protein; cAMP, adenosine 3′,5′-cyclic monophosphate; CAT, catalase; Chol, cholesterol; ChREBP, carbohydrate element response binding protein; CPT1, carnitine palmitoyl transferase 1; CREB, cAMP-response element binding protein; Cyp7a1, cholesterol 7*α*-hydroxylase 1; DCR, 2,4-dienoyl-CoA reductase; DGAT2, diacylglycerol O-acyltransferase 2; ERK, extracellular-signal-regulated kinase; eWAT, white adipose tissue e=epididymal; FA, fatty acid; FAS/Fasn, fatty acid synthase; FCR, fractional catabolic rates; FFA, free fatty acid; FGF, fibroblast growth factor; FGFR4, FGF receptor 4; FOS, fructo-oligosaccharides; FXR, farnesoid X receptor; G6Pase, glucose-6-phosphatase; GA, galacturonic acid; GK, glucokinase; Gpam, glycerol- 3-phosphate acyltransferase, mitochondria; GPAT, glycerol 3-phosphate acyltransferase; GS, glycogen synthase; GSH, glutathione; GSH-Px, glutathione peroxidase; GSK-3*β*, glycogen synthase kinase-3*β*; HDL-C, high density lipoprotein-cholesterol; HE, high esterification; HFD, high-fat diet; HFP, high-fat pectin; HM, high methylation degree; HMG-CoA, *β*-hydroxy *β*-methylglutaryl-CoA; HOMA-IR, homeostatic model assessment for insulin resistance; HPPS, haw pectin pentasaccharide; IRS-1, insulin receptor substrate-1; JNK, c-Jun N-terminal kinase; L/A, leptin to adiponectin ratio; LDL, low-density lipoprotein; LDL-C, low-density lipoprotein-cholesterol; Lepr, leptin receptor; LFP, low-fat pectin; LKB1, liver kinase B1; LM, low methylation degree; Lpl, lipoprotein lipase gene; MCP-1, monocyte
chemoattractant protein-1; MDA, malondialdehyde; Mlxipl, MLX interacting protein-like gene; MUPA, mono-unsaturated fatty acids; NEFA, non-esterified fatty acid; NFR-1, nuclear
respiratory factors-1; NF-*κ*B, nuclear factor-*κ*B; Nrf2, nuclear factor (erythroid-derived 2)-like 2; pAkt, phosphorylated protein kinase B; pAMPK, phosphorylated AMP-activated protein kinase; PAP, phosphatidate phosphohydrolase; PEPCK, phosphoenolpyruvate carboxykinase; PGC-1*α*, peroxisome proliferator-activated receptor-*γ* coactivator 1 alpha; pGK,
phosphorylated glucokinase; pGS, phosphorylated glycogen synthase; pGSK-3*β*, phosphorylated glycogen synthase kinase-3*β*; pIRS-1, phosphorylated insulin receptor substrate-1; PKA,
protein kinase A; Pnpla2, patatin-like phospholipase domain containing 2 gene; PPAR*α*, peroxisome proliferator-activated receptor alpha; PPAR*γ*, peroxisome proliferator activated
receptor gamma; Prkaa2, AMP-activated protein kinase, alpha 2 catalytic subunit gene; PUFA, polyunsaturated fatty acids; PYY, peptide YY; RXR, retinoid X receptor; Scd1, stearoyl-CoA
desaturase 1; SCFA, short-chain fatty acids; SFA, saturated fatty acids; SHP1, short heterodimer partner 1; Slc2a4, solute carrier family 2 (facilitated glucose transporter), member 4; SOD,
superoxide dismutase; SRC, Src tyrosine kinase; Srebf1, sterol regulatory element binding transcription factor 1; STAT3, signal transducer and activator of transcription 3; TAC, total
antioxidation capacity; TBARS, thiobarbituric acid reactive substances; TC, total cholesterol; TG, triglyceride; TNF-*α*, tumor necrosis factor a; UNK, unknown; VLDL, very-low density
lipoprotein; W, week.

**Table 3 nutrients-15-00157-t003:** Changes induced by a pectin-enriched diet to address liver injury in alcoholic liver disease. Summary of metabolic changes induced by pectin in studies addressing the improvement of liver damage in alcoholic liver injury.

Animal Species	Type of Pectin	Pectin Amount % or g/day/kg	Duration in Days and (weeks)	Liver Steatosis Lipid Metabolism	Plasma Lipids Plasma Metabolites	ALT Liver Inflammation Liver Metabolism	Bile Acid and Cholesterol Metabolism or Metabolites SCFA	IM Composition and Gut Homeostasis	Ref
C57BL/6J mice, female	Apple pectinHM (73%)	6.5%	Prevention 21 days (3 w)	↓ steatosis ↓ TG		↓ **ALT** ↓ liver weight ↓ liver TNFα, IL1β, IL6, CCL2, TGFβ	↓ fecal BA, TUDCA & UDCA,	↑ Bacteroides, proportion of Enterobacteriaceae ↓ reduced IM diversity ↑ goblet cells, Reg3β (colon & Ileum), reg3γ (Ileum)	[81]
C57BL/6J mice, female	Apple pectin	0.4%, 1%, 2% and 6.5%	Curative 7 days (1 w) in 28 days (4 w)	↓ steatosis ↓ TG	not modify alcohol absorption	↓ ALT, CCL2, CCL3, TNF, IL1β ↑ bacterial genes involved in carbohydrate, lipid, and amino-acid metabolism	↓ Tryptophan, Indole ↑ total AhR agonists	↑ Bacteroides, Bacteroidetes, Lactobacillus ↓ Firmicutes ↑ Proteobacteria and Enterobacteriaceae (6.5% pectin) ↑ Reg3β, reg3γ(colon & Ileum) (2% & 6.5% pectin)↑ Cyp1a1, AhRr, il17, il22(colon) (2% & 6.5% pectin)	[75]
C57BL/6J mice, female	Apple pectin	6.5%	Prevention 21 days (3 w)	↓ TG ↓ BAT UCP1		↓ ALT ↓ liver TNFα, IL1β, CCR2, CCL2, CCL3	↓ plasma total BA ↓ plasma CA, MCAβ, MCAω, DCA, TCA, TMCA ↓ liver MCAβ, TDCA ↑ liver TCDCA ↑ caecum CA, CDCA, UDCA, TCA, TMCA, TUDCA ↓ caecum MCAω, DCA, LCA, TCDCA, TDCA ↓ ileum MRP2, SGLT1, Glut2, CD36, Fabp1 mRNA ↑ ileum MRP3 mRNA ↑ colon ASBT, OST mRNA	↑ Bacteroides, Enterobacteriacae ↓ Lactobacillus and Enterococcus	[111]
C57BL/6J mice, female	Apple pectin (PE) vs. PE + B. fragilis ATCC25285 (BFPE)	2%	Prevention 10 days (1.4 w)	↓ steatosis and neutrophil infiltration ↓ TG ↓ IL-1α, IL-1β, and TNF-α, CD36, PPARγ mRNA, ↓ Liver TLR4 mRNA	↓ plasma LPS, LBP, IL-2, IL-12	↓ ALT (BFPE)	↑ acetate (BFPE), propioniate, butyrate in cecal contents ↑ IPA & IAA & Tryptophan (BFPE), ILA in colon contents	↑ shannon index ↑ Bacteroides, B. fragilis, Bacteroidetes, Bacteroidales, Proteobacteria, Enterobacterales, Escherichia-Shigella, Lachnospirales ↓ Firmicutes, Erysipelotrichales, Monoglobales, Peptostreptococcales-Tissierellales, Dubosiella, Monoglobus, Allobaculum, Faecalibaculum, Romboutsia ↑ colon goblet cell counts, MUC2 mRNA ↑ colon ZO-1(BFPE), IL-22, Reg3β, and Reg3γ	[112]

Abbreviations: AhR, aryl hydrocarbon receptor; ALT, alanine aminotransferase; ASBT, apical sodium-dependent bile acid transporter; BA, bile acid; BAT, brown adipose tissue; BFPE, apple pectin and *B. fragilis* ATCC25285; CA, cholic acid; CCL, CC chemokine ligand; CCR2, C-C chemokine receptor type 2; CDCA, cheno-deoxycholic acid; Cyp1a1, cholesterol 1*α*-hydroxylase 1; DCA, deoxycholic acid; Fabp1, fatty acid binding protein1; Glu2, glucose transporter 2; HM, high methylation degree; IAA, indoleacetic acid; IL-1*β*, interleukin 1 Beta; IL-6, interleukin 6; IL-17, interleukin 17; IL-22, interleukin 22; ILA, indolelactic acid; IPA, indole-3 propioniate; LBP, LPS-binding protein; LCA, lithocholic acid; LPS, lipopolysaccharide; MCA, muricholic acids; MRP, multidrug-resistance-associated protein; MUC2, mucin 2; OST, organic solute and steroid transporter; PPAR*γ*, peroxisome proliferator activated receptor gamma; Reg, regenerating family member; SGLT1, sodium/glucose cotransporter 1; TCA, tauro-cholic acid; TCDCA, tauro-chenodeoxycholic acid; TDCA, tauro-deoxycholic acid; TG, triglyceride; TGF-*β*, transforming growth factor beta; TLR, toll-like receptor; TMCA, tauro-muricholic acid; TNF-*α*, tumor necrosis factor *α*; TUDCA, tauro-ursodeoxycholic acid; UCP1, uncoupling protein 1; UDCA, ursodeoxycholic acid; UDCA, urso-deoxycholic acid; ZO-1, zonula occludens-1.

## Data Availability

Not applicable.

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
