# Peer review of "Pectin in Metabolic Liver Disease"

_nutrients, 2022, doi:10.3390/nu15010157_

Round 1

Reviewer 1 Report

Please see the attached file for comments.

Author Response

We thank you for your careful reading of our review. We corrected all our typing errors and answered all the points, except the one concerning the presentation of legends and abbreviations of tables, because the guidelines of Nutrients give some different information concerning the format. However, if the editorial board further suggests modifying this point, we will do this.

Point 1:

Over the last two decades, many criticisms have been voiced about the nomenclature and definition of non-alcoholic fatty liver disease (NAFLD) in regards not only to the prominent role that alcohol plays in the definition but also to the negative impacts of the nomenclature including trivialization, stigmatization and less consideration of the disease in health policy. Recently, a consensus by an international panel of experts recommended a change in name for NAFLD to metabolic (dysfunction) associated fatty liver disease (MAFLD). This issue is reviewed by Fouad, Y., Waked, I., Bollipo, S., Gomaa, A., Ajlouni, Y., & Attia, D., 2020 (What's in a name? Renaming ‘NAFLD’ to ‘MAFLD’." Liver international 40.6: 1254-1261). For details please see also a paper by Eslam, M., Sanyal, A. J., George, J., Sanyal, A., Neuschwander-Tetri, B., Tiribelli, C., ... & Younossi, Z., 2020 (MAFLD: a consensus-driven proposed nomenclature for metabolic associated fatty liver disease. Gastroenterology, 158(7), 1999-2014). Therefore, I suggest that the authors take into account the above comments regarding the nomenclature of the described liver disease. It should be noted that the authors used the acronym “MAFLD” once (in the title of section 3, line 221), but it is not explained anywhere in the text.

We thank you for raising this important point.

We totally agree with your point of view and at the beginning, we wanted to use MAFLD instead of NAFLD. However, after a though literature review the term NAFLD/NASH is still the preferred terminology by the scientific, academic, and medical communities. Moreover, in all of the studies cited concerning the effects of pectin, the authors used the term NAFLD/NASH. Therefore, in order to avoid any misunderstanding and to render our review easier to access outside the NAFLD/MAFLD specialists we decided to use the term NAFLD.

However, as we agree that it is important to emphasize the new terminology, we included in the introduction paragraph a phrase and referenced the paper of Eslam et al, 2020:

“Of note, a consensus of international experts proposed that the disease acronym be changed from NAFLD to metabolic (dysfunction) associated fatty liver disease or ‘MAFLD’ to more accurately reflect pathogenesis and better integrate a current understanding of patient heterogeneity 5. However, as the studies cited in the present review mostly used the term NAFLD, we will use this acronym across the present work.”

Points 2-37

We thank you for highlighting these inconsistencies and we corrected all of them in the revised version of our manuscript. 

Reviewer 2 Report

"Pectin in metabolic liver disease" is a review that wanted to highlight the role of a soluble fiber such as pectin, in the treatment of alcohol-dependent and non-alcohol dependent liver failure.

The authors describe many works in which the usefulness of pectin is highlighted especially in laboratory animals and conclude that there are too few clinical trials in humans to definitively decree the efficacy of pectin in the treatment of ALD, NAFLD or NASH.

The review is clear and easy to read and the organization of the paragraphs is well done. However, a few suggestions could further improve the review.

First of all, at the end of the Introduction, the reason why the authors chose to evaluate the efficacy of pectin in the bibliography is reported in two lines: I would suggest explaining with a few more details the reasons why pectin, and not other substances, has been selected for this review. Furthermore, a small presentation of what will be reported, also to ignite the reader's curiosity.

On p. 2, L. 82: I would add a reference.

Tables 2 and 3 are important and full of news. Since each of these takes up a whole page, I would suggest writing the description in the upper part of the table, while in the lower part, instead of listing all the abbreviations, I would refer you to the paragraph on abbreviations at the end of the work.

The figures are clear, explanatory and clearly summarize the mechanisms described in the text. I would suggest starting to mention them even earlier in the text. The suggestion about abbreviations also applies here.

Author Response

We thank you for your careful reading of our review. We corrected all our typing errors and answered all the points, except the one concerning the presentation of legends and abbreviations of tables, because the guidelines of Nutrients give some different information concerning the format. However, if the editorial board further suggests modifying this point, we will do this. 

Point 1: First of all, at the end of the Introduction, the reason why the authors chose to evaluate the efficacy of pectin in the bibliography is reported in two lines: \I would suggest explaining with a few more details the reasons why pectin, and not other substances, has been selected for this review. Furthermore, a small presentation of what will be reported, also to ignite the reader's curiosity.

We added a phrase to better explain our choice and referenced three additional references about the use of inulin:

“The use of different fibers, such as inulin, in the context of metabolic diseases and more particularly metabolic liver diseases was addressed by several studies16-18. Among the different fibers available, pectin is a soluble fiber found in different fruits and vegetables. It modulates the gut microbiome and in recent studies, it showed promising results in protecting the liver from metabolic injuries such as alcohol and western diet. This review will provide an overview on pectin biological effect with an emphasis on its role as a microbiome editing strategy and its potential role in modulating metabolic liver disease.”

Point 2: On p. 2, L. 82: I would add a reference.

We add the reference which is also cited previously in the text.

Point 3: Tables 2 and 3 are important and full of news. Since each of these takes up a whole page, I would suggest writing the description in the upper part of the table, while in the lower part, instead of listing all the abbreviations, I would refer you to the paragraph on abbreviations at the end of the work.

As we follow the guidelines of Nutrients, we did not move the abbreviations, but as suggestions of reviewer 1, we homogenized the Title and legend presentation.

Point 4: The figures are clear, explanatory and clearly summarize the mechanisms described in the text. I would suggest starting to mention them even earlier in the text. The suggestion about abbreviations also applies here.

According to these remarks, we mention figure 2 earlier in the text, after the first paragraph concerning the use of pectin in high-fat diet. For figure 3, we mention it at the end of the first paragraph concerning the use of pectin in alcoholic liver disease. We moved the figures in the text for a better reading of the manuscript. As for point 3, we did not move the abbreviations. However, if the editorial service support further gives the possibility to bypass Nutrients guidelines, we will further change the place of the abbreviations. If we could do that, the place of figures 2 and 3 will be also improved, so we specifically discuss this point in the cover letter.